# Analysis and Optimization of Mechanical Properties of Laser-Sintered Cellulose/PLA Mixture

**DOI:** 10.3390/ma14040750

**Published:** 2021-02-05

**Authors:** Hui Zhang, David L. Bourell, Yanling Guo

**Affiliations:** 1College of Mechanical and Electrical Engineering, Northeast Forestry University, Harbin 150040, China; zhanghui@nefu.edu.cn; 2Department of Mechanical Engineering, The University of Texas at Austin, Austin, TX 78712, USA; dbourell@mail.utexas.edu

**Keywords:** cellulose/PLA mixture, laser sintering, process parameter optimization, thermal post-processing, mechanical properties

## Abstract

This studied aimed at improving the mechanical properties for a new biopolymer feedstock using laser-sintering technology, especially when its laser-sintered parts are intended to be applied in the industrial and medical fields. Process parameter optimization and thermal post-processing are two approaches proposed in this work to improve the mechanical properties of laser-sintered 10 wt % cellulose-polylactic acid (10%-CPLA) parts. Laser-sintering experiments using 2^3^ full factorial design method were conducted to assess the effects of process parameters on parts’ mechanical properties. A simulation of laser-energy distribution was carried out using Matlab to evaluate the experimental results. The characterization of mechanical properties, crystallinity, microstructure, and porosity of laser-sintered 10%-CPLA parts after thermal post-processing of different annealing temperatures was performed to analyze the influence of thermal post-processing on part properties. Image analysis of fracture surfaces was used to obtain the porosity of laser-sintered 10%-CPLA parts. Results showed that the optimized process parameters for mechanical properties of laser-sintered 10%-CPLA parts were laser power 27 W, scan speed 1600 mm/s, and scan spacing 0.1 mm. Thermal post-processing at 110 °C produced best properties for laser-sintered 10%-CPLA parts.

## 1. Introduction

Laser-sintering (LS) technique is well disposed to produce complex structural parts with tailored properties by additive manufacturing [1,2]. Compared to traditional subtractive manufacturing, this additive manufacturing technique offers many advantages such as functional integration, elimination of tooling and fixtures, rapid fabrication of customized products in small batches, and high materials’ usage rate [3,4,5]. With these benefits, LS positively facilitates the development of various manufacturing industries combined with creative design concept and sustainability considerations [6,7,8]. Thus, development of eco-friendly, cost-effective, and high-performance feedstock becomes one of the research focuses for the advancement of LS technology [9,10,11], which highlights the importance of polymeric materials derived from natural resources [12].

Polylactic acid (PLA) shows tremendous potential to replace the petroleum-derived polymers for industrial applications, which is bio-sourced, biodegradable, and biocompatible [13,14]. PLA is not only processed into products via traditional extrusion, injection molding, and thermoforming [15,16], but also has been commercially utilized as the feedstock of an additive manufacturing technique called fused deposition modelling (FDM) in recent years [17,18,19,20]. However, up to now, LS of PLA has not been well developed, with limited reports [21,22,23]. This thorny problem can be attributed to unavailable powder-shaped PLA, unstable bonding performance of raw material, and unsatisfied mechanical properties and dimensional accuracy.

A fully degradable polymeric mixture consisting of commercial polylactic acid (PLA) powder and cellulose powder was prepared as a potential LS feedstock in our previous study [24]. The effects of filler loading on the performance of laser-sintered α-cellulose/PLA parts were investigated. Results showed that increasing cellulose loading improved dimensional accuracy but reduced mechanical strength of laser-sintered parts. The mixture with 10 wt% cellulose added (abbreviated as 10%-CPLA) was considered to be a promising feedstock for LS in terms of balanced performance. Sticking to the purpose of sustainable development of LS technology, this study continues to focus on improving part mechanical properties to fabricate laser-sintered 10%-CPLA parts more suitable for high-performance applications. In addition to material design, adjusting build parameters during LS processing and post-processing are two other methods to customize and optimize the performance of LS parts.

The 2^3^ full factorial experimentation including eight factorial runs and two center points was adopted to investigate the correlation between three processing parameters (laser power, scanning speed, scan spacing) on the resulting mechanical strength. A simulation of the distribution of laser energy under the influence of process parameters was used to verify the results obtained from the full factorial experiments. Process parameter optimization is an effective method to improve part mechanical properties, but high laser-energy density may reduce dimensional accuracy. Excess laser energy causes the surrounding powder to stick to the part, which has a detrimental effect for hollow part fabrication such as topological parts and parts with tiny holes.

To solve this problem, thermal post-processing was proposed to improve the mechanical properties of laser-sintered parts fabricated using low laser energy. Thermal post-processing was carried out at different temperatures for laser-sintered 10%-CPLA parts. The mechanical strength, crystallinity, microstructure, and porosity of laser-sintered 10%-CPLA parts were characterized to evaluate the influence of thermal post-processing on part properties. Specifically, the porosity was evaluated by calculating the area percentage of voids from scanning electron microscope (SEM) images of sample fracture surfaces based on an image analysis method.

## 2. Materials and Methods

### 2.1. Material Preparation

PLA Ingeo^TM^ 3001D was supplied in pellet form by NatureWorks LLC (Blair, NE, USA) and crushed cryogenically into powders by eSUN (Shenzhen, China). Powder apparent density was 0.62 g/cm^3^. PLA Ingeo^TM^ 3001D consisted of poly L-lactic acid (PLLA) with a D-lactide content of 1.4%. The particle size was in the range of 176–296 μm. Cellulose powder was purchased from Aladdin Industrial Corporation (Shanghai, China), with size varying from 25 to 65 μm.

As shown in Figure 1, dried cellulose fiber and PLA Ingeo^TM^ 3001D powder were mechanically mixed in a high-speed mixer at a mass ratio of 10/90. Its glass transition temperature (*T*_g_) of 58 °C, two crystallization peaks at 89 °C and 151 °C, and the melting temperature (*T*_m_) of 167 °C were obtained using an STA449F3 simultaneous thermal analyzer (Netzsch Group, Selb, Germany) at a heating rate of 10 °C/min, and the thermal degradation temperature (*T*_d_) of 358 °C was obtained by a thermogravimetric analyzer (Discovery TGA 5500, New Castle, DE, USA) at a heating rate of 10 °C/min.

### 2.2. Full Factorial Laser-Sintering Experimentation

In order to reduce laser power output, the layer thickness and preheating temperature were almost set to extremum values. The layer thickness during laser-sintering powder spreading was 0.25 mm, according to the size of 10%-CPLA particles, which was larger than most particles and made the accuracy of laser-sintered parts as high as possible. Meanwhile, 145 °C was chosen as the preheating temperature, near but below the agglomeration temperature of the 10%-CPLA mixture. Since the layer thickness was set as a fixed value, laser power, scan speed, and scan spacing related to the laser-energy density (two-dimensional surface), as seen as Equation (1) [25], were considered as the three factors to investigate their influence on mechanical properties of laser-sintered 10%-CPLA parts. Appropriate processing windows were laser power 23–27 W, scan speed 1600–2000 mm/s, and scan spacing 0.1–0.2 mm. The full factorial scheme was established, as shown in Table 1. To predict the mechanical properties via a regression model, an analytic method based on multiple regression and residues (AMMRR) method was performed to analyze the significant influence of variables including laser power, scan spacing, scan speed, and their second-order factors on the tensile strength and flexural strength. LS experiments were conducted on an AFS-360 rapid prototyping machine produced by Longyuan AFS Co., Ltd. (Beijing, Hebei Sheng, China), which was equipped with a CO_2_ laser. The maximum laser power output was 55 W, and the laser beam diameter was approximately 0.4 mm. Laser-energy density *L*_D_ (W∙(mm^2^/s)^−1^) was calculated using:(1)$$L_{D}=\frac{P}{v\;\cdot \;d}$$
where *P* is the laser power (W), *v* is scan speed (mm/s), and *d* is scan spacing (mm).

### 2.3. Laser-Energy Distribution

The Gaussian distributed surface (2D) heat flux has been widely accepted as the distribution of laser beam energy in LS process modelling [26], represented as Equation (2) [27]. The laser-energy output is strongest in the center of laser spot and dissipates moving toward the edge.
(2)$$q\left(x,y\right)=\frac{2P}{\pi \omega^{2}}\exp\left(-\frac{2r{\left(x,y\right)}^{2}}{\omega^{2}}\right)$$
where *q*(*x,y*) is the heat flux intensity, *P* is the laser power, *ω* is the radius of the laser spot, 0.2 mm, and *r*(*x,y*) is the radial distance from the laser beam center.

A CO_2_ laser operates in a pulsed or continuous manner. If the laser energy is seen as a continuous form, the energy density distribution in cross section for a single laser line scan can be calculated as Equation (3) [28].
(3)$$E\left(y\right)=\sqrt{\frac{2}{\pi }}\cdot \frac{P}{wv}\cdot \exp\left(-\frac{2y^{2}}{\omega^{2}}\right)$$
where *v* is scan speed of laser beam.

Practically, laser scanning is a successive process during LS. The overlapping scan lines will lead to a linearly overlapped laser-energy deposition, shown as Figure 2. When the scan spacing is expressed as *d*, if one scanning line is set as *y* = 0, the Nth scan line after the first scan line is *y* = *Nd*. The distance between a certain point *Q* (*x,y*) and the *N*th scan line will be *y*-*Nd*. Thus, the thermal effect of the *N*th scan line on the point *Q* (*x,y*) is shown as Equation (4).
(4)$$E\left(y\right)=\sqrt{\frac{2}{\pi }}\cdot \frac{P}{\omega v}\cdot \exp\left[-\frac{2{\left(y-Nd\right)}^{2}}{\omega^{2}}\right].$$

The stack laser energy (*E*_s_(*y*)) for these *N* scan lines is calculated as Equation (5). Based on this formula, Matlab 2016 was utilized to analyze the effects of process parameters on *E*_s_(*y*) visually in this study.
(5)$$E_{s}\left(y\right)=\overset{n}{\underset{N=0}{\sum }}\left\{\sqrt{\frac{2}{\pi }}\cdot \frac{p}{\omega v}\cdot \exp\left[-\frac{2{\left(y-Nd\right)}^{2}}{\omega^{2}}\right]\right\}$$

### 2.4. Thermal Post-Processing

The annealing temperature should be higher than the *T*_g_ of 10%-CPLA because the motion of molecular chains only starts above *T*_g_. Besides, it is also required that the annealing temperature should be lower than *T*_m_ to avoid the LS parts being remelted. Experiments demonstrated samples were softened obviously when exposed to heat at 140 °C, which was not conductive to keep the shape accuracy of laser-sintered parts of complex structure. Based on these two principles and experiment phenomenon, the thermal post-processing temperatures were chosen as 65 °C, 100 °C, 110 °C, 120 °C, and 130 °C, respectively. The samples, fabricated with low laser-energy density input (process parameters of laser power 23 W, scan speed 2000 mm/s, and scan spacing 0.2 mm), were heated for 24 h and then cooled to room temperature in the thermostat.

### 2.5. Characterization of the Performance of Laser-Sintered 10%-CPLA

The phase composition of the 10%-CPLA feedstock and its laser-sintered parts was identified by X-ray diffraction (XRD) (X’ Pert 3 Powder, Malvern Instruments, Malven, UK). The testing conditions were scattering angles (2*θ*) 5–50° and a scan rate of 8°/min.

Microscopic imaging of LS sample cross sections was performed using a COXEM EM-30 (COXEM., Co., Ltd., Daejeon, Korea) scanning electron microscope (SEM). The samples were treated by gold sputtering prior to SEM observation. The quality of part surface appearance was evaluated using an electron microscope.

To analyze the porosity of laser-sintered 10%-CPLA parts, their cross-sectional SEM pictures were first transformed into binary images using image analysis to enhance the contrast between voids and solid material. Subsequently, the percentage of black area representing the voids in every binary image was estimated based on the Matlab 2016 bwarea algorithm.

The density of the laser-sintered 10%-CPLA parts, *ρ* (g/cm^3^), was confirmed according to Equation (6) after measuring the flexural sample weight and dimensions of three sides.
(6)$$\rho =\frac{m}{l_{1}\;\cdot \;l_{2}\;\cdot \;l_{3}}$$
where *m* was the mass of a laser-sintered part (g), and *l*_1_, *l*_2_, and *l*_3_ were the length (cm), width (cm), and height (cm), respectively.

Tensile testing of laser-sintered 10%-CPLA parts was implemented according to ASTM D638-14 at room temperature in a universal electromechanical testing machine (Byes 3003, Bangyi Precision Measurement (Shanghai) Co., Ltd., Shanghai, China) at a crosshead speed of 5 mm/min. Standard dog bone-shaped testing samples were printed with dimensions of 166 mm × 13 mm × 3.2 mm. Additionally, ASTM D790-17 3-point bending method was used to test the flexural strength of thin specimens of dimensions 127 mm × 12.7 mm × 3.2 mm. The support span was 60 mm, crosshead speed was 2 mm/min, and midspan deflection was 15 mm. Both tensile strength and flexural strength were expressed by averaging the testing values of nine repeated measurements.

## 3. Results and Discussions

### 3.1. Optimization of Process Parameters

The laser energy, density, and mechanical properties responding to the full factorial design scheme are listed in Table 1. Generally, the density and mechanical properties of laser-sintered 10%-CPLA parts were increased and strengthened as the laser-energy density increased.

AMMRR results, including but not limited to normal plotting of the standardized effects and Pareto chart of the standardized effects (Figure 3), revealed that the significance effect degree of the three process parameters for tensile strength was scan spacing > laser power > scan speed > (scan spacing × scan speed), and for flexural strength was scan spacing > scan speed > laser power > (scan spacing × scan speed).

Insignificant terms, including (laser power × scan speed) and (laser power × scan spacing), were excluded to improve the accuracy of regression models. Thus, an empirical relationship between tensile strength, flexural strength, and process parameters can be expressed by Equations (7) and (8), respectively, with *A*-laser power (W), *B*-scan speed (mm/s), and *C*-scan spacing (mm).
*Y*_T_ = −7.1649 + 0.7854*A* + 0.0091*B* + 63.565*C* − 0.0983*BC*(7)
*Y*_F_ = 11.2828 + 0.7047*A* + 0.0077*B* + 53*C* − 0.1053*BC*(8)

Laser power 27 W, scan speed 1800 mm/s, and scan spacing 0.15 mm were used to validate the math models’ reliability. According to Equations (7) and (8), the fitting value of tensile strength and flexural strength under this process parameter regime was 13.36 MPa and 23.61 MPa and the 95% confidence interval was (12.89, 13.82), (22.54, 24.68), respectively. Actual tensile strength of 12.90 MPa and flexural strength of 22.98 MPa obtained by LS experiment was consistent with the regression equations. The optimized process parameters for mechanical properties of laser-sintered 10%-CPLA parts were identified to be laser power 27 W, scan speed 1600 mm/s, and scan spacing 0.1 mm.

### 3.2. The Distribution of Stack Laser Energy

Figure 4 depicts the distribution of *E*_s_(*y*) under the influence of different process parameters. Clearly, increasing laser power and decreasing scan spacing and scan speed led to large maximum values of *E*_s_(*y*). Besides, the change in *E*_s_(*y*) seemed more sensitive to scan spacing than to laser power and scan speed when comparing Figure 4a–c. This phenomenon had a good agreement with the results obtained from AMMRR analysis above. Also of note, scan spacing not only played a vital role in the maximum value of *E*_s_(*y*) but also affected the uniformity of distribution of *E*_s_(*y*). When the scan spacing was greater than or equal to the radius of laser spot, there were peaks and troughs in the *E*_s_(*y*) curve, which would also cause an uneven quality of laser-sintered parts.

### 3.3. The Effects of Thermal Post-Processing on the Properties of Laser-Sintered 10%-CPLA Parts

#### 3.3.1. Mechanical Properties

The results in Figure 5 indicate that both mechanical properties and the density of the parts were improved after different post-processing anneals. When the temperature of post-processing grew to 110 °C, the green part tensile property increased from 2.48 MPa to the maximum value of 4.36 MPa, flexural strength was strengthened from 10.40 MPa to the maximum value of 12.16 MPa, and the density (the colored bar in Figure 5) slightly increased from 0.919 g/cm^3^ to 0.958 g/cm^3^. When the temperature of post-processing reached 130 °C, the tensile strength and flexural strength, respectively, dropped to 3.01 MPa and 10.78 MPa. Meanwhile, the density decreased to 0.93 g/cm^3^. Thus, the best process-processing temperature for mechanical properties was 110 °C.

#### 3.3.2. Microstructures and Porosity

The surface quality of laser-sintered 10%-CPLA parts under different thermal post-processing conditions is depicted in Figure 6. It shows that both the size and number of pores in the parts decreased and the surface became smoother with higher transparency after post-processing. These phenomena imply that the microstructure, porosity, and even the crystallinity of laser-sintered 10%-CPLA parts might have changed.

SEM images of part fracture surfaces magnified by 60 times were used to evaluate an overall microstructure of the cross section, depicted as Figure 7. SEM images of part fracture surface magnified by 150 times were used to characterize the area between two layers. Image analysis (Figure 8) derived from the SEM micrographs shown in Figure 7 were obtained to calculate the porosity of the green parts and the parts after thermal post-processing.

As observed in Figure 7a and Figure 8a, green parts fabricated using low laser energy were porous with voids of varying sizes. Highly irregular voids larger than 400 μm in size were observed in the interlaminar junction areas, which indicates poor bonding between layers. However, thermal post-processing reduced part porosity. That is, not only the size but also the number of voids were decreased, as illustrated in Figure 7b–f and Figure 8b–f. These defects, caused by insufficient melting of PLA particles during LS processing, were remedied since the restarted thermal motion of the polymer segments during thermal post-processing resulted in an improved bonding strength between layers and particles. In addition, annealing treatment would help to remove the residual stress that existed in laser-sintered 10%-CPLA parts, reducing the shrinkage distortion of PLA particles during the condensation process and bringing in a denser internal structure.

Additionally, the porosity of the green parts and parts after thermal post-processing was described quantitatively in Table 2. The results also indicate that part porosity dropped after thermal post-processing and the minimum value was located at an annealing temperature of 110 °C. Besides, the data verified that the porosity between two subsequent layers was much higher than that within a layer, comparing the porosity of SEM images magnified 150 times and that of SEM images magnified 60 times.

#### 3.3.3. Thermal Properties and Crystallinity

XRD curves of cellulose, PLA 3001D, and 10%-CPLA mixtures are displayed in Figure 9a. It can be seen from the XRD plot for cellulose that a wide peak and a strong peak were observed near 14.9°~16.7° and 22.8°, respectively, which corresponded to (101), (101(-)), and (002) crystal planes for cellulose Ι. PLA 3001D had three diffraction peaks at 2*θ* = 16.8°, 21.6°, and 23.9°, which corresponded to (200/110), (204), and (016) crystal planes of α crystal in PLA, respectively. The XRD pattern of 10%-PLA was almost consistent with the XRD curve of PLA 3001D. The XRD curves of laser-sintered 10%-CPLA parts and the parts after post-processing are shown in Figure 9b. These curves show a feature different from the XRD curve of the 10%-CPLA mixture. All these five XRD curves had a quite distinct peak at 2*θ* = 16.9°, representing (200/110) crystal planes of PLA, a small peak at 2*θ* = 19.5° corresponding to the characteristic (203) crystal plane of PLA, and a very small peak at 2*θ* = 22.5°, which might be associated with (002) crystal planes for cellulose. This means the crystallization of both PLA 3001D and cellulose occurred during LS processing of 10%-CPLA.

Comparing the six curves in Figure 9b, results revealed an increase in the degree of crystallinity of laser-sintered 10%-CPLA parts after thermal post-processing, which reached the highest value when the annealing temperature was 110 °C. Higher crystallinity means a more regular arrangement of internal structures in 10%-CPLA, leading to improved tensile strength and flexural strength.

To sum up, thermal post-processing was considered to have a positive effect on the mechanical properties of laser-sintered 10%-CPLA parts, reducing the porosity of laser-sintered 10%-CPLA parts, improving the flatness of parts’ surface and increasing the crystallinity of laser-sintered parts. Results verified the best performance of laser-sintered parts was located at the annealing temperature of 110 °C. Process parameters’ optimization is a straightforward way to improve the mechanical properties of solid parts, but optimized process parameters with high laser-energy density may cause poor definition and accuracy of laser-sintered hollow parts such as the scaffold illustrated in Figure 10a. However, thermal post-processing is effective in improving the mechanical properties of hollow parts, which are printed using low laser-energy density and have a better dimensional accuracy, as shown in Figure 10b.

## 4. Conclusions

The influence of process parameter optimization during LS processing and thermal post-processing at different temperatures of laser-sintered parts on the mechanical properties of laser-sintered 10%-CPLA parts were studied. Specific results of this work are listed as follows.

When the layer thickness was 0.25 mm and the preheating temperature was 145 °C, the ranking of influential factors on tensile strength of laser-sintered 10%-CPLA parts from highest to the lowest was scan spacing > laser power > scan speed > (scan spacing × scan speed), and for flexural strength was scan spacing > scan speed > laser power > (scan spacing × scan speed). Scan spacing had the most significant impact on the mechanical properties of laser-sintered 10%-CPLA composites.

Regression equations were obtained to quantitatively describe the relationship between tensile strength, flexural strength, and the process parameters (laser power, scan spacing, and scan speed), which could predict the mechanical properties of laser-sintered 10%-CPLA parts fabricated using specific process parameters. Furthermore, the optimized process parameters for mechanical properties of laser-sintered 10%-CPLA parts were laser power 27 W, scan speed 1600 mm/s, and scan spacing 0.1 mm.

Thermal post-processing with the annealing temperature from 65 °C to 130 °C improved the surface quality, density, and crystallinity of laser-sintered 10%-CPLA parts, leading to improved mechanical properties. Results revealed that 110 °C was the best annealing temperature to improve the general performance of laser-sintered 10%-CPLA parts.

## Figures and Tables

**Figure 1 materials-14-00750-f001:**
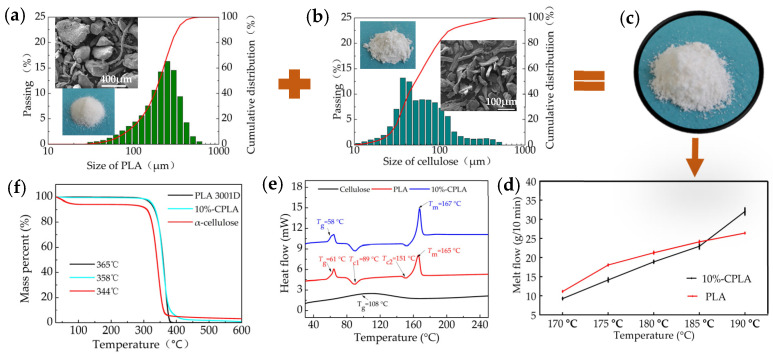
Preparation of 10%-CPLA mixture: (**a**) characteristics of PLA 3001D; (**b**) characteristics of cellulose; (**c**) the appearance of 10%-CPLA; (**d**) melt flow of PLA 3001D and 10%-CPLA; (**e**) DSC curves of 10%-CPLA and its components; (**f**) TGA curves of 10%-CPLA and its components.

**Figure 2 materials-14-00750-f002:**
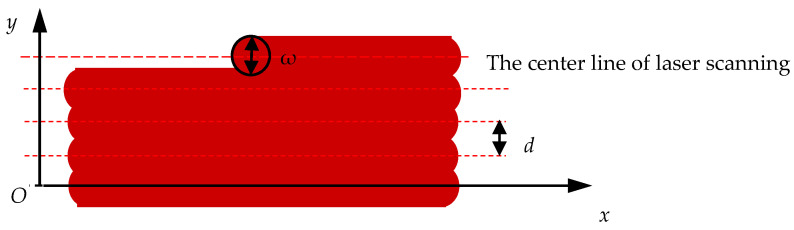
A schematic of the scanning track of laser spot in LS processing.

**Figure 3 materials-14-00750-f003:**
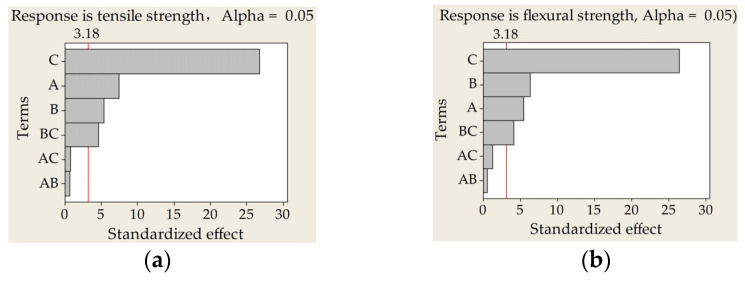
Pareto charts of the standardized effects on mechanical properties of laser-sintered 10%-CPLA parts: (**a**) standardized effect of tensile strength; (**b**) standardized effect of flexural strength. (A, laser powder; B, scan speed; C, scan spacing).

**Figure 4 materials-14-00750-f004:**
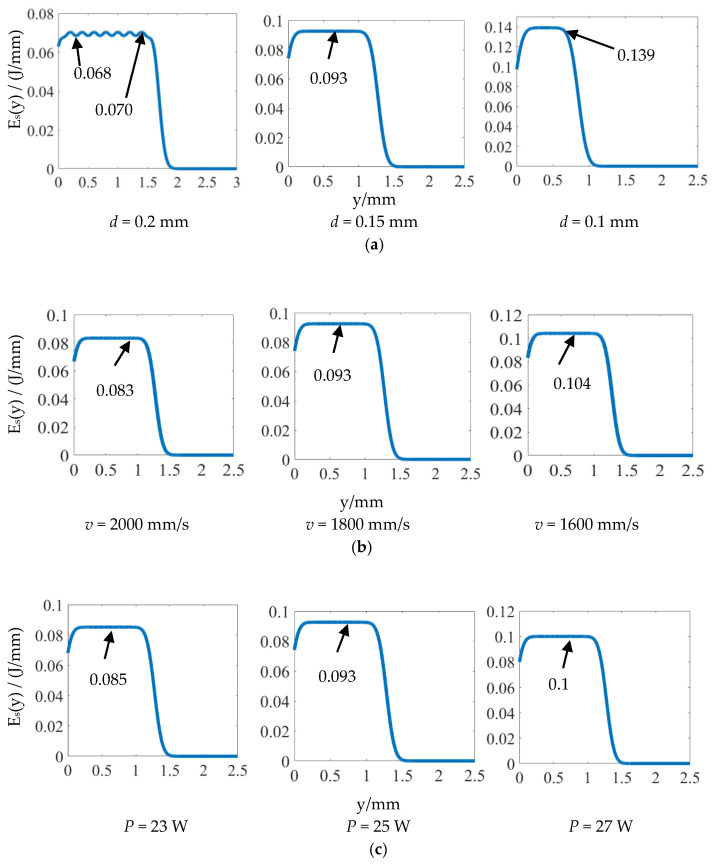
The effects of process parameters on the distribution of laser energy (*E*_s_(*y*), J/mm): (**a**) scan spacing (*d*) is variable, *P* = 25 W, *v* = 1800 mm/s; (**b**) scan speed (*v*) is variable, *P* = 25 W, *d* = 0.15 mm; (**c**) laser power (*P*) is variable, *v* = 1800 mm/s, *d* = 0.15 mm.

**Figure 5 materials-14-00750-f005:**
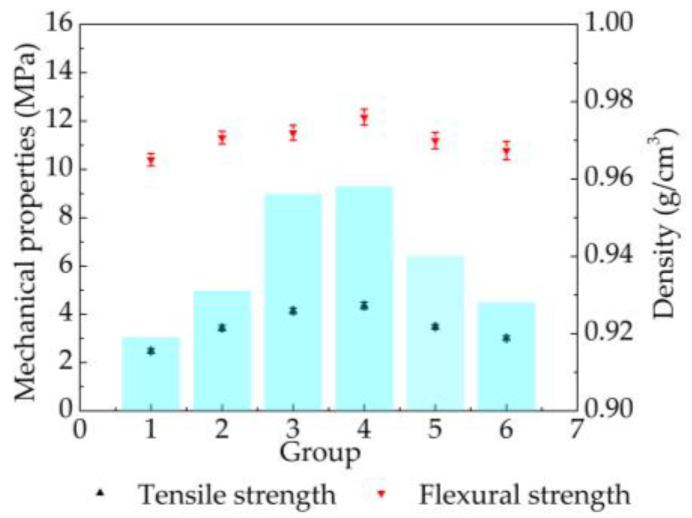
Mechanical properties and density of laser-sintered 10%-CPLA parts under different thermal post-processing (1, the green part; 2, 65 °C; 3, 100 °C; 4, 110 °C; 5, 120 °C; 6, 130 °C).

**Figure 6 materials-14-00750-f006:**
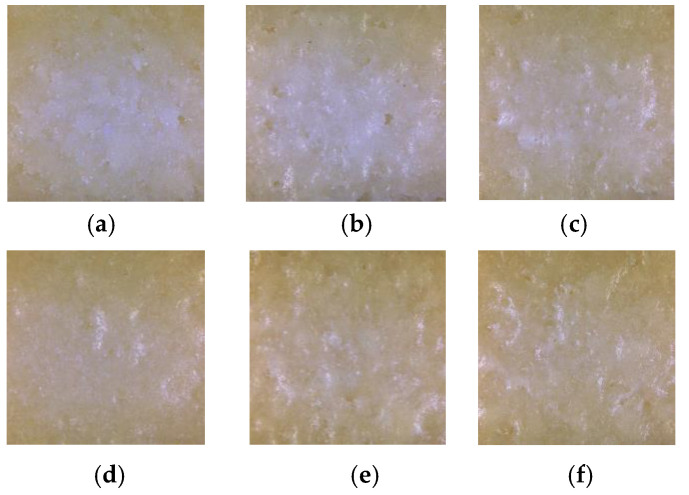
The surface of laser-sintered 10%-CPLA parts: (**a**) the green part; (**b**) 65 °C; (**c**) 100 °C; (**d**) 110 °C; (**e**) 120 °C; (**f**) 130 °C.

**Figure 7 materials-14-00750-f007:**
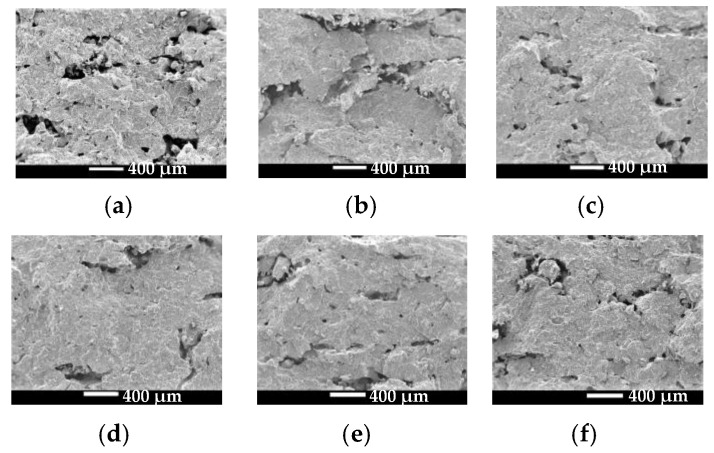
SEM images of the fracture surface of the green part and the parts after thermal post-processing: (**a**) the green part; (**b**) 65 °C; (**c**) 100 °C; (**d**) 110 °C; (**e**) 120 °C; (**f**) 130 °C.

**Figure 8 materials-14-00750-f008:**
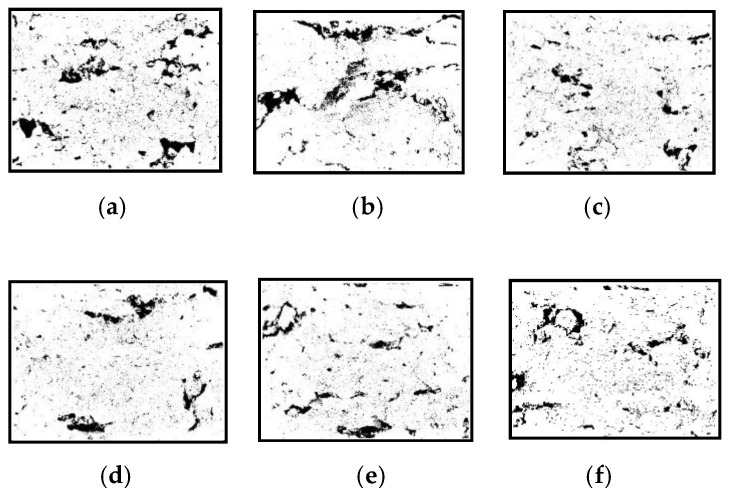
Image analysis images of corresponding fracture surfaces of laser-sintered 10%-CPLA parts: (**a**) the green part; (**b**) 65 °C; (**c**) 100 °C; (**d**) 110 °C; (**e**) 120 °C; (**f**) 130 °C.

**Figure 9 materials-14-00750-f009:**
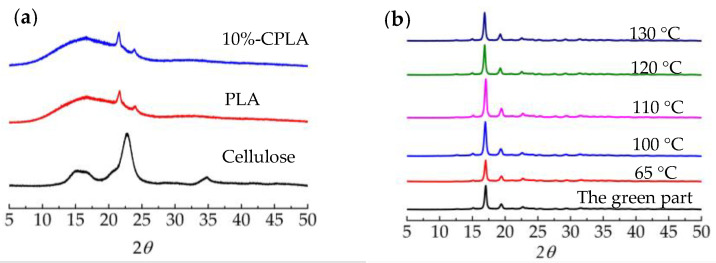
XRD curves of feedstock and laser-sintered 10%-CPLA parts: (**a**) XRD curves of cellulose, PLA 3001D, and 10%-CPLA mixture; (**b**) XRD curves of laser-sintered 10%-CPLA parts.

**Figure 10 materials-14-00750-f010:**
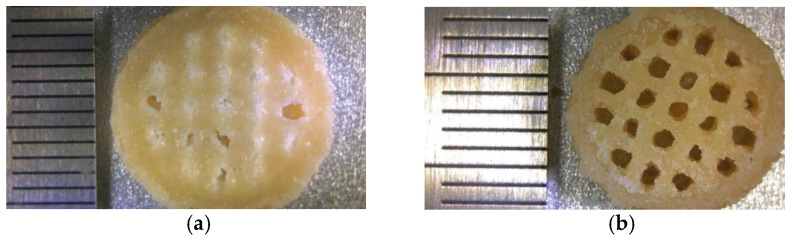
The laser-sintered scaffold of 10%-CPLA: (**a**) under best process parameters; (**b**) via thermal post-processing along with low laser-energy input.

**Table 1 materials-14-00750-t001:** Full factorial design scheme and mechanical properties of laser-sintered 10%-CPLA parts.

NO.	Process Parameters			Laser- Energy Density (×10^4^ W/(m^2^/s))	Density (g/cm^3^)	Tensile Strength (MPa)	Flexural Strength (MPa)
	Laser Power (W)	Scan Speed (mm/s)	Scan Spacing (mm)				
1	23	1600	0.1	143.75	1.14	16.13	28.25
2	27	1600	0.1	168.75	1.18	19.09	30.95
3	23	2000	0.1	115	1.11	15.95	27.65
4	27	2000	0.1	135	1.15	18.66	29.25
5	23	1600	0.2	71.88	1.00	6.29	16.86
6	27	1600	0.2	84.38	1.03	10.17	19.25
7	23	2000	0.2	57.5	0.92	2.48	10.4
8	27	2000	0.2	67.5	1.03	5.51	14.98
9	25	1800	0.15	92.59	1.10	10.83	22.57
10	25	1800	0.15	92.59	1.11	11.19	22.63

**Table 2 materials-14-00750-t002:** The porosity of 10%-CPLA green parts and parts after thermal post-processing.

Group		1	2	3	4	5	6
The annealing temperature (°C)		The green part	65	100	110	120	130
Porosity (%)	×60 times	10.49	10.01	7.09	6.72	7.69	8.25
	×150 times	26.80	23.54	21.34	17.66	20.77	22.85

## Data Availability

No new data were created or analyzed in this study. Data sharing is not applicable to this article.

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
