# Peer review of "Analysis and Optimization of Mechanical Properties of Laser-Sintered Cellulose/PLA Mixture"

_materials, 2021, doi:10.3390/ma14040750_

Round 1

Reviewer 1 Report

The authors aiming at improving the mechanical properties of laser-sintered 10%-CPLA parts, approaching two methods: process parameter optimization during LS processing and thermal post-processing of laser-sintered parts.

The study is easy to follow and covers an hot topic, but some minor issues should be improved before publication. Several typos should be corrected thorough the text.

Abstract and introduction section: Please better describe the aim of the work.

Conclusion Section: This paragraph required a general revision to eliminate redundant sentences and to add some "take-home message".

Author Response

Comments 1: Abstract and introduction section: Please better describe the aim of the work.

Response 1: Thanks for your valuable suggestion. I revised the whole text and corrected the wrong typos. Besides, the abstract and introduction section was modified to explain the aim of work, and the aim of the methods. ‘Aiming at improving the mechanical properties for a new biopolymer feedstock using for laser sintering technology, especially when its laser-sintered parts are intended to be applied in the industrial and medical fields.’ was presented in the abstract. ‘Sticking to the purpose of sustainable development of LS technology, this study continues to focus on improving part mechanical properties to fabricate laser-sintered 10%-CPLA parts more suitable for high-performance applications. In addition to material design, adjusting built parameters during LS processing and post-processing are two other methods to customize and optimize the performance of LS parts.’ was presented in the introdution.

Comments 2: Conclusion Section: This paragraph required a general revision to eliminate redundant sentences and to add some "take-home message".

Response 2: Thanks for your advice. The conclusion has been revised. The first paragraph was condensed, and some important effect rule and data were listed as follows:

The influence of process parameter optimization during LS processing and thermal post-processing at different temperatures of laser-sintered parts on the mechanical properties of laser-sintered 10%-CPLA parts were studied. Specific results of this work are listed as follows:

When the layer thickness was 0.25 mm and the preheating temperature was 145 ℃, the ranking of influential factors on tensile strength of laser-sintered 10%-CPLA parts from highest to the lowest was scan spacing > laser power > scan speed > (scan spacing × scan speed), and for flexural strength was scan spacing > scan speed > laser power > (scan spacing × scan speed). Scan spacing had the most significant impact on the mechanical properties of laser-sintered 10%-CPLA composites.

Regression equations were obtained to quantitatively describe the relationship between tensile strength, flexural strength and the process parameters (laser power, scan spacing and scan speed), which could predict the mechanical properties of laser-sintered 10%-CPLA parts fabricated using specific process parameters. Furthermore, the optimized process parameters for mechanical properties of laser-sintered 10%-CPLA parts were laser power 27 W, scan speed 1600 mm/s, and scan spacing 0.1 mm.

Thermal post-processing with the annealing temperature from 65 ℃ to 130 ℃ improved the surface quality, density and crystallinity of laser-sintered 10%-CPLA parts, leading to improved mechanical properties. Results revealed that 110 ℃ was the best annealing temperature to improve the general performance of laser-sintered 10%-CPLA parts.

Reviewer 2 Report

Following their previous study, the authors aim at enhancing mechanical properties of laser-sintered 10%-CPLA parts through studying the effects of processing parameters scanning speed, scan spacing and laser power and applying thermal post-processing. While the topic is of interest for community in the armor research field, the authors need to address the following major comments:

  • It is not clear when you talk about optimal material in page 4 line 56. We do not have an optimal material because you can always use other materials which might give better results.
  • There are typos and grammatical issues in the manuscript. Authors need to read it carefully and make sure there are no remaining such issues. For example:

Page 1: However, up to now, LS of PLA hasn’t been

Correct: However, up to now, LS of PLA has not been

Figures such as Figure 1 should come after the paragraph referring to them not before that.

  • Page 2: Its glass transition temperature (Tg) was 58 ℃, two crystallization peaks were observed at 89 ℃ and 151 ℃, the melting temperature (Tm) was 167 ℃ and the thermal degradation temperature (Td) was 358 ℃.

Have you done DMA to get these data? You need to mention details of characterization tools and their method.

Does your material have shape memory effect? Have you thought to change the glass transition temperature of the material by changing the composition for specific applications such as 4D printing?

Furthermore, reducing Tg will result in using less energy for thermal post-processing. As a future study you might consider changing the Tg for these purposes.

  • residual analysis (AMMRR) method

It is not clear why the method is called AMMRR and how it is calculated. You need to explain these.

It is also not clear what is alpha and A, B, C, AC, AB and BC in Fig 3.

It is also unclear how data in Fig 3 are obtained, what is standardized effects and Pareto chart. You need to explain these.

  • Page 6: comparing Figure 3 (a), Figure 3 (b) and Figure 3 (c).

These should be Figure 4 (a), …

  • Figure 10 is not referred to in the text before it and rather only in conclusion!
  • How did you come up with this as appropriate?

Appropriate processing windows were laser power 23-27 W, scan speed 1600-2000 mm/s and scan spacing 0.1-0.2 mm.

  • Page 3: Where q(x,y) should be where q(x,y)
  • Page 3: shown as Figure 2

as shown in Figure 2

  • Page 4: The samples, fabricated with process parameters of laser power 23 W, scan speed 2000 mm/s and scan spacing 0.2 mm

Based on what, you chose these samples for thermal post-processing? Why these samples and why not samples with other processing conditions?

  • Page 4: dimensions of three sides: (change dot to colon)

  • Page 4: ‘Where m was the mass’ to ‘where m was …’

  • Page 4: respectively, (cm). remove (cm)

  • Page 4: Tensile testing of laser-sintered 10%-CPLA parts was implemented according to ASTM D638-14 at room temperature in a universal electromechanical testing machine (Byes 3003) at a crosshead speed of 5 mm/min. Standard dog-bone-shaped testing samples were printed with dimensions of 166 mm × 13 mm × 3.2 mm.

How many samples did you test to make sure repeatability? You need to report standard deviation of data. This is specially important for the discussion related to Figure 5, because the difference between the tensile data is not large to conclude that the observed trend justifies presence of a maximum point. Also you have only one point after the maximum. It will be more convincing if there are more points after the maximum. Can you do more test beyond the fifth point?

  • Page 5: Figure 4 depicts

  • Page 7: When the temperature of post-processing reached 120 ℃, the tensile strength and flexural strength respectively dropped to 3.48 MPa and 11.19 MPa.

You think what the reason for this is, with increasing from 110 ℃ to 120 ℃, the trend is reversed?

  • Figure 5: What is the meaning of the coloured bar chart? You have two set of data for tensile and flexural strengths, however it is not clear what are the bars.

  • Page 7: Image analysis images (Figure 8) derived

Image analysis (Figure 8) derived

  • Page 8: It was improved after post-processing in that

Sentence needs re-writing

  • Page 8: 3.3.2. Thermal properties and crystallinity

It should be 3.3.3

  • Figure 9 should after the paragraph referring to it not before.
  • Conclusions: listed as follows:
  • References are not consistent: For example, ref 1title has all words with capital while ref 2 only first word is in capital

Author Response

Dear reviewer,

    Thank you for all the comments and suggestions. The point-to-point responses were expressed in the attached document.

Reviewer 3 Report

This manuscript can be considered for publication in the present form after modifying the title a bit as given below: 

cellulose/PLA parts should be replaced by cellulose/PLA composites. 

Author Response

Dear reviewer,

  Thanks for your suggestion. I changed the title 'Analysis and optimization of mechanical properties of laser-sintered cellulose/PLA parts' as 'Analysis and optimization of mechanical properties of laser-sintered cellulose/PLA mixture' according to your suggestion. Mixture means the material is obtained by mixing the ingredients mechanically, so I think it is more suitable to describe 10%-CPLA I studied.

Reviewer 4 Report

The present paper entitled “Analysis and Optimization of Mechanical properties of laser-sintered cellulose/PLA parts” by the authors Hui Zhang, David L. Bourell, and Yanling Guo studies the impact of different processing parameters in the mechanical properties of a laser-sintered 10wt% cellulose-polylactic acid mixture. Results revealed that scan spacing has the greater influence on tensile and flexural strength of the parts. Thermal post-processing techniques were also studied showing an improvement of the surface quality, density and crystallinity of the laser-sintered parts.

After careful evaluation of the manuscript, I consider that it would typically appeal to the readers of Materials, and therefore I would recommend this paper for publication, although there are some aspects which should be clarified and where changes may help the reader:

  • The design of experiments presented by authors has not been properly planned. It is necessary to justify the choice of factors, levels and ranges employed, in addition with the response variables selected. For that it is convenient a more detailed statistical analysis including additional graphs (normal probability plots of each factor, Interaction effects matrix plot, residual graphs, etc) that could be included in the supporting information of the manuscript.
  • Based on that more meaningful analysis authors should justify why they have maintained constant in their model two relevant factors such as laser thickness and preheating temperature.
  • The design cannot be named as full factorial unless it would include 3^k trials when k factors are considered, each at three levels.
  • Authors should also consider the possibility of including replication in their model which will improve the reliability of the estimates.
  • The equations (1 to 5) presented should be properly cited, or explained if they have been deduced. 

Author Response

Dear reviewer,

  Thanks for your careful review. The point-to-point responses to comments and suggestions were expressed in the attached document.

Round 2

Reviewer 4 Report

It seems that all comments and suggestions have been correctly addressed, and the statistical analysis included looks complete and well-conducted. However, I disagree with authors about naming the design conducted as full factorial. As authors have stated in their response a full factorial design can also be a two-level factor. That brings a total of 2^3=8 trials for three factors which shall result in a total of 10 runs if two additional center points are added. However, authors have studied three levels for each factor, namely 23 W, 25 W and 27 W for Laser power; 1600 mm/s, 1800 mm/s and 2000 mm/s for scan speed; and 0.1 mm, 0.15 mm and 0.2 mm for scan spacing. Therefore, a total of 3^3=27 trials are necessary in order to have a full factorial design. Please see Anderson, M.J., and Whitcomb, P. J., RSM Simplified: Optimizing Processes using Response Surface Methods for Design of Experiments. New York, NY: CRC Press, 2005. I really encourage authors to change any reference to full factorial in the manuscript and explain the type of design that they have included instead.

Author Response

Dear reviewer,

  Thank you for the comments. The detailed responses are in the attached document.
